# In Vivo Bioluminescence Imaging of HBV Replicating Hepatocytes Allows for the Monitoring of Anti-Viral Immunity

**DOI:** 10.3390/v13112273

**Published:** 2021-11-13

**Authors:** Katrin Manske, Annika Schneider, Chunkyu Ko, Percy A. Knolle, Katja Steiger, Ulrike Protzer, Dirk Wohlleber

**Affiliations:** 1Institute of Molecular Immunology, Klinikum Rechts der Isar, TUM School of Medicine, Technical University of Munich, 80333 Munich, Germany; katrin.manske@tum.de (K.M.); annika.schneider@tum.de (A.S.); percy.knolle@tum.de (P.A.K.); 2Institute of Virology, Technical University of Munich, 81675 Munich, Germany; ckko@krict.re.kr (C.K.); protzer@tum.de (U.P.); 3Infectious Diseases Therapeutic Research Center, Therapeutics & Biotechnology Division, Korea Research Institute of Chemical Technology (KRICT), Daejeon 34114, Korea; 4German Center for Infection Research (DZIF), Munich Partner Site, 81675 Munich, Germany; 5Institute of Pathology, Technical University of Munich, 81675 Munich, Germany; katja.steiger@tum.de

**Keywords:** liver immunology, immunity, CD8 T cells, T cell dysfunction, bioluminescence imaging (BLI), HBV

## Abstract

Immunity against hepatitis B virus (HBV) infection is complex and not entirely understood so far, including the decisive factors leading to the development of chronic hepatitis B. This lack of a mechanistic understanding of HBV-specific immunity is also caused by a limited number of suitable animal models. Here, we describe the generation of a recombinant adenovirus expressing an HBV 1.3-overlength genome linked to luciferase (Ad-HBV-Luc) allowing for precise analysis of the quantity of infected hepatocytes. This enables sensitive and close-meshed monitoring of HBV-specific CD8 T cells and the onset of anti-viral immunity in mice. A high dose of Ad-HBV-Luc developed into chronic hepatitis B accompanied by dysfunctional CD8 T cells characterized by high expression of PD1 and TOX and low expression of KLRG1 and GzmB. In contrast, a low dose of Ad-HBV-Luc infection resulted in acute hepatitis with CD8 T cell-mediated elimination of HBV-replicating hepatocytes associated with elevated sALT levels and increased numbers of cytotoxic HBV-specific CD8 T cells. Thus, the infectious dose was a critical factor to induce either acute self-limited or chronic HBV infection in mice. Taken together, the new Ad-HBV-Luc vector will allow for highly sensitive and time-resolved analysis of HBV-specific immune responses during acute and chronic infection.

## 1. Introduction

Hepatitis B virus (HBV) infections are still considered a major global health problem, accounting for 296 million chronically infected people worldwide in 2019 [1]. However, HBV infection of adults results in viral persistence in only 5% of cases [2,3,4]. The decisive factors determining chronic disease outcome are currently still poorly understood and may also be associated with the organ of infection, the liver, and its role in regulating immune responses. The liver is regarded as a lymphoid organ, which is unique in its microenvironment and cell composition [5,6]. Hence, in the liver, the induction of anti-viral T cell immunity as well as the induction of tolerance mechanism may take place depending on the total amount and quality of immune stimulation during the infection [7,8,9,10]. 

Chronic hepatitis B is characterized by weak and progressively exhausted HBV-specific CD8 T cells or their Bim-mediated deletion [11,12,13,14]. Conventional anti-viral therapy leads to elimination of HBV in only 1% of patients due to its poor induction of CD8 T cell immunity, being crucial for the elimination of infected hepatocytes [15,16,17,18,19,20,21]. In chronic HBV carriers, HBV-specific CD8 T cells are reported to be limited in their proliferative capacity as well as their potency to secrete anti-viral cytokines, such as IFNγ [11]. Moreover, these HBV-specific T cells co-express markers, e.g., PD1, CTLA-4, Tim-3 and 2B4, that are associated with a dysfunctional phenotype [9]. Notably, the dysfunctional phenotype is described to be even more pronounced on intrahepatic HBV-specific T cells compared to their peripheral counterparts [22]. 

In the meantime, the importance of HBV-specific CD8 T cells for the elimination of HBV infection is widely accepted. Hence, the development of novel treatment options against chronic hepatitis B aims to restore HBV immunity, in particular HBV-specific CD8 T cells. Therefore, a deeper understanding of the differences in the induction of HBV-specific CD8 T cell immunity during acute self-limited and chronic HBV infection is required. However, suitable and sensitive infection models are rare because of HBV’s species-specificity [23,24]. Mice are not naturally permissive for HBV, and appropriate small animal models to analyze HBV-specific immune responses are scarce [25]. Thus, we aimed to develop a preclinical model system for HBV infection in mice that allows one to compare immune responses during acute self-limited and chronic infection using the same model. Previous model systems are based, e.g., on the hydrodynamic injection of HBV-DNA into mice, leading to acute disease outcomes [26]. In contrast, chronic HBV infection is established, e.g., by the transfer of HBV genomes via adeno-associated viral vectors (AAV) [27]. However, it is not recommended to compare HBV-specific immune responses using different model systems in mice, resulting in either acute or chronic HBV infection.

Here, we used a recombinant replication-deficient adenoviral vector to transfer HBV 1.3-overlength genomes into murine hepatocytes. In addition, we linked the HBV DNA to a sequence encoding the click beetle green (CBG99) luciferase reporter protein. This enabled highly sensitive real-time quantification of viral infection in the liver and thereby, indirectly, anti-viral immunity by bioluminescence imaging. We were able to induce either acute self-limited or chronic HBV infection in mice using the same vector but different infectious doses. Analyzing intrahepatic HBV-specific T cells during acute self-limited and chronic Ad-HBV-Luc infection, we demonstrate that the kinetics of virus elimination correlate strongly with either cytotoxic or exhausted HBV-specific CD8 T cell phenotype, respectively.

## 2. Materials and Methods

### 2.1. Adenoviral Vector

We designed a pENTR4 vector (Thermo Fisher Scientific, Waltham, MA, USA) for cloning into replication-deficient recombinant adenovirus serotype 5 (Thermo Fisher Scientific, USA), which consists of a HBV 1.3-overlength genome [28] linked to a click beetle green (CBG99) luciferase separated by a P2A linker site derived from the porcine teschovirus. The P2A linker and the CBG99-luciferase were cloned in frame onto the second open reading frame coding for the HBc protein and after the HBV polyA-sequence. Generation and titration of serotype 5 adenoviruses was performed as described before [29].

### 2.2. Mice

C57Bl/6 mice were purchased from Charles River (Sulzfeld, Germany) and Cor93 TCR-transgenic mice (B6.Cg-Ptprca Pepcb Tg(TcraBC10,TcrbBC10)3Chi/J) were purchased from The Jackson Laboratory (Bar Harbour, ME, USA). All mice were maintained under specific pathogen-free (SPF) conditions in the central animal facility of the Klinikum rechts der Isar, Technical University of Munich according to the guidelines of the Federation of Laboratory Animal Science Association. Male mice older than 6 weeks were used. 

Adenoviral vectors were diluted in 0.9% sodium chloride solution and injected intravenously (i.v.) into the tail vein in a total volume of 100 µL.

Cell depletion was performed by means of weekly i.v. injection of 30 μg anti-mouse CD8α (clone 2.43, BioXCell, Lebanon, NH, USA) diluted in PBS.

For adoptive T cell transfer, naïve CD8 T cells were isolated from spleen and lymph nodes of Cor93 TCR-transgenic mice by negative magnetic bead separation (Miltenyi Biotec, Bergisch Gladbach, Germany). A total of 10,000 CD45.1 T cells were injected i.v. in 100 µL PBS into the tail vein.

### 2.3. Bioluminescence Imaging

Mice were anesthetized with 2.5% isoflurane (CP-Pharma, Burgdorf, Germany) and injected i.p. with 100 mg/kg bodyweight D-luciferin-K-salt (PerkinElmer, Waltham, MA, USA) in 100 µL of PBS 5 min before bioluminescence imaging. In vivo bioluminescence imaging was performed with an IVIS Lumina LT-Series III instrument (PerkinElmer LAS, Germany). Analysis of the bioluminescence signal per liver was performed by Living Image^®^ 4.5 software (PerkinElmer, Waltham, MA, USA), determining photons per second in the region of the liver.

### 2.4. Alanine Aminotransferase (ALT) Measurement

For the detection of liver damage according to serum ALT (sALT) levels, blood was sampled from the Vena facialis of mice. Fresh whole blood was analyzed by Reflotron^®^ Plus system and Reflotron^®^ GPT (ALT) tests (Hoffmann-La Roche, Basel, Switzerland).

### 2.5. Quantitative Real-Time PCR

Real-time PCR for adenoviral or HBV genomes was performed with 100 ng genomic DNA isolated from 5 mg liver tissue by NucleoSpin^®^ Tissue kit (Macherey-Nagel, Düren, Germany). RNA was isolated using NucleoSpin RNA Kit (Macherey-Nagel, Düren, Germany). cDNA was synthesized from 1 µg RNA using the SensiFAST cDNA Synthesis Kit (Bioline Reagents, London, UK) according to the manufacturer’s instructions. Cyclophilin PCR was performed to normalize HBV pgRNA PCR. 

PCR was performed with Takyon No Rox SYBR MasterMix dTTP Blue Kits (Eurogentec, Seraing, Belgium).

The following primers were used at a final concentration of 100 nM:
(1)Adenovirus DNAforward (TAAGCGACGGATGTGG)reverse (CCACGTAAACGGTCAAAG); (2)HBV DNAforward (GTTGCCCGTTTGTCCTCTAATTC)reverse (GGAGGGATACATAGAGGTTCCTTGA); (3)HBV pgRNAforward (GAGTGTGGATTCGCACTCC)reverse (GAGGCGAGGGAGTTCTTCT); (4)Cyclophilinforward (ATGGTCAACCCCACCGTGT) reverse (TTCTGCTGTCTTTGGAACTTTGTC).

Initial denaturation was performed at 95 °C for 10 min, followed by 45× cycles of denaturation (95 °C, 20 s), annealing (60 °C, 10 s) and extension (72 °C, 60 s) and a melting curve (65–95 °C, 0.11 °C/s).

PCR reactions were performed with the LightCycler^®^ 480 II System (Roche Molecular Systems, Pleasanton, CA, USA) and data were analyzed by LightCycler^®^ 480 SW 1.5.1 software.

### 2.6. Quantification of HBe and HBs Antigen in Blood

Blood was sampled from the Vena facialis of infected mice. Plasma was isolated using Microvette^®^ 500 LH-gel (Sarstedt, Nürnbrecht, Germany) and 5 min of centrifugation at 10,000× *g*. Plasma was diluted in PBS and analyzed by ARCHITECT i1000SR System and ARCHITECT HBeAg or HBsAg Reagent Kits (Abbott Laboratories, Irving, TX, USA).

### 2.7. Isolation of Liver-Associated Lymphocytes

For the isolation of lymphocytes from livers, mice were sacrificed and the liver was perfused via the portal vein with PBS, mechanically dissected and passed through a sieve. After washing in PBS, liver cells were incubated in Gey’s balanced-salt solution (PAN Biotech, Aidenbach, Germany) and 0.125 U/mg collagenase type 2 (Worthington Biochemical Corporation, Lakewood, NJ, USA) for 10 min at 37 °C. Lymphocytes were isolated by Percoll (GE Healthcare, Chicago, IL, USA) gradient centrifugation. Therefore, the cell pellet was resolved in 40% Percoll solution and under layered with 80% Percoll solution. After gradient centrifugation, lymphocytes were washed in PBS and used directly for flow cytometric analysis, restimulation or the xCELLigence kill assay.

### 2.8. Isolation of PBMCs

During the isolation of PBMCs, 250 to 600 µL of heparinized whole blood was transferred to 15 mL sample tubes and filled up to 3 mL with PBS. The blood–PBS layer was under-layered with 3 mL of Pancoll (PAN-Biotech GmbH, Aidenbach, Germany). After gradient centrifugation (800× *g*, 20 min), lymphocytes were washed in PBS and used directly for flow cytometric analysis. 

### 2.9. Flow Cytometry

Surface staining was performed at 4 °C in PBS (+1% FCS/1% BSA). After staining with Cor93-multimers 40 min on ice, all other surface antibodies were incubated for a further 20 min. Cells were fixed for 30 min at room temperature using IC fixation buffer (Invitrogen, Waltham, MA, USA). Intranuclear staining was performed according to manufacturer’s instructions (True-NuclearTM Transcription Factor Staining Buffer Set, BD Biosciences, Franklin Lakes, NJ, USA). 

Endogenous or transferred CD45.1 Cor93-specific T cells were stained by H-2Kb/MGLKFRQL multimers specific for the HBcAg (93–100) epitope (SP/7397-03, ProImmune Ltd., Littlemore, UK, dilution 1:10) for 40 min on ice. For phenotypic analysis, the following antibodies were used: anti-CD8α (clone KT15, dilution 1:200), anti-CD45.1 (clone A20, dilution 1:200), anti-CD44 (clone IM7, dilution 1:200), anti-PD1 (clone 29F.1A12, dilution 1:200), anti-KLRG-1 (clone 2F1/KLRG1, dilution 1:100), anti-TNF (MP6-XT22, dilution 1:200), anti-IFNγ (clone XMG1.2, dilution 1:200), anti-Granzyme B (clone GB11, dilution 1:100), anti-TIGIT (clone 4D4, dilution 1:200), anti-TOX (clone TXRX10, dilution 1:200), and fixable viability dye eFluorTM 780 (dilution 1:1000). Antibodies were purchased from ProImmune (Sarasota, FL, USA), Sony Biotechnology (Bothell, WA, USA), eBioscience/Invitrogen (Waltham, MA, USA) or BioLegend Inc. (San Diego, CA, USA). 

For ex vivo restimulation and intracellular cytokine staining, lymphocytes were isolated from livers and incubated for 6 h with 100 nM MGLKFRQL peptide or 100 nM SIINFEKL peptide (JPT, Berlin, Germany) and 3 µg/mL brefeldin A (eBioscience/Invitrogen, Waltham, MA, USA) at 37 °C. After staining with anti-CD8α and viability dye eFluorTM 780, CD8 T cells were fixed in IC fixation buffer (eBioscience, USA) for 20 min at room temperature. Intracellular staining of TNF and IFNγ was performed in permeabilization buffer. Intracellular staining was performed using True-NuclearTM Transcription Factor Staining Buffer Set (BD Biosciences, Franklin Lakes, NJ, USA). 

Flow cytometric analysis was performed with SP6800 Spectral Analyzer (Sony Biotechnology, Bothell, WA, USA). The data were analyzed by FlowJo software V10.1 (TreeStar Inc., Ashland, OR, USA).

### 2.10. Northern Blot Analysis of HBV RNA

Northern blot analysis was conducted as described previously [30]. Briefly, total RNA from mouse liver tissues (ca. 20 mg) was extracted using TRIzol reagent (Ambion Corporation, Naugatuck, CT, USA). The concentration of RNA was measured by NanoDrop (Thermo Scientific, Waltham, MA, USA) and 12 µg of intrahepatic RNA was electrophoresed on an agarose gel containing formaldehyde. RNA was then transferred to a positively charged nylon membrane through capillary action and cross-linked to the membrane by UV irradiation. Prior to the capillary transfer, ribosomal RNA species (18S and 28S) were visualized by ethidium bromide staining for loading control. The membrane was hybridized with a 3 kb digoxigenin-labelled HBV DNA probe followed by HBV RNA visualization with DIG luminescence detection kit (Hoffmann-La Roche, Basel, Switzerland) according to the manufacturer’s instruction. For the probe generation, digoxigenin coupled to dUTP were incorporated during the PCR reaction with a template plasmid DNA, and the primers as follows: pCH-9/3091 plasmid (HBV genotype D), HBV89-F (5′-TTCTAGATACCGCCTCAGCTCT-3′) and HBV3090-R (5′-TGGTGCGCAGACCAATTTAT-3′) [31].

### 2.11. Histology

Mouse livers were fixed in 4% neutral-buffered formalin solution for 48 h, buffered in PBS, dehydrated (Leica Biosystems, Nussloch,, Germany) and embedded in paraffin. Serial 2 µm-thin sections were prepared with a rotary microtome (HM355S, Thermo Scientific, Waltham, MA, USA) and subjected to histological and immune-histochemical analysis. Hematoxylin–eosin (HE) staining was performed on deparaffinized sections with eosin and Mayer’s haemalaun according to the standard protocol. For immunohistochemistry, slides were deparaffinized and pretreated with Epitope Retrieval solution 1 for 30 min. Staining was performed using the BondMax RXm system (Leica Biosystems, Nussloch, Germany, all reagents from Leica) with a primary antibody against the HBcore protein (RB-1413, 1:50 in antibody diluent, Invitrogen/Thermo Scientific, Waltham, MA, USA). Bound antibody was detected with a Polymer Refine detection kit (Leica Biosystems, Nussloch, Germany) without a post primary reagent and visualized with DAB as a dark brown precipitate. Counterstaining was performed with hematoxylin.

### 2.12. Real-Time Viability: xCELLigence Kill Assay

The xCELLigence system (Real-Time Cell Analysis (RTCA) System, ACEA Biosciences Inc., San Diego, CA, USA) was used to determine the cell-viability of target cells during co-culture experiments. Primary murine hepatocytes were isolated from non-infected C57Bl/6 mice as described previously [32]. A total of 1 × 10^4^ primary murine hepatocytes were seeded per well of a E-96-wellplate coated with 0.02%, collagen R (#47254.02, SERVA Electrophoresis GmbH, Heidelberg, Germany). Then, 24 h after attachment, hepatocytes were infected with Ad-HBV1.3 (MOI 5) or left untreated and then cultured for a further 48 h. Thereafter, primary murine hepatocytes were co-cultured with 5 × 10^5^ CD8 T cells from the livers of mice infected either 10^7^ pfu or 10^8^ pfu Ad-HBV-Luc. Intrahepatic CD8 T cells were isolated by Percoll gradient followed by magnetic bead separation (#130-096-543, Miltenyi Biotec, Bergisch Gladbach, Germany). Cell viability is illustrated as the cell index normalized to the start of co-culture.

### 2.13. Statistics

The Mann–Whitney-test was performed for statistical analysis. Analysis was performed by Prism 7 software, version 7.0c (GraphPad, San Diego, CA, USA). Significance was set at *p* < 0.05 and denoted as * *p* < 0.05, ** *p* < 0.01, *** *p* < 0.001, **** *p* < 0.0001 and ns = not significant. All results are expressed as mean ± standard deviation (SD).

## 3. Results

### 3.1. Transfer of HBV Genomes in Mice

HBV is described as strongly species-specific and infects only humans and chimpanzees. Therefore, mice are not naturally permissive for HBV infection. To study the immune response against HBV in mice, we developed an adenoviral shuttle system transferring HBV 1.3-overlength genomes into murine hepatocytes. We linked the second truncated HBVcore antigen (HBcAg) sequence to a click beetle green luciferase (CBG99) gene separated by a P2A site, ensuring expression under the viral HBc promoter (Ad-HBV-Luc, Figure 1A).

Bioluminescence imaging of mice infected with Ad-HBV-Luc showed a distinct bioluminescent signal at the liver area, in contrast with either uninfected or Ad-HBV1.3-infected mice, where the adenoviral vector encodes only for the HBV 1.3-overlength genome (Figure 1B). Quantification of circulating HBeAg, a secretory form of the HBc protein used for the clinical quantification of HBV, in blood and the bioluminescence signals detected in the liver of Ad-HBV-Luc-infected mice showed direct correlation of the two markers with increasing infectious doses (Figure 1C). Therefore, we concluded that in vivo bioluminescence imaging allows for the close-meshed quantification of viral load in the liver. Moreover, bioluminescence correlated directly with HBeAg levels in serum. Notably, while antigen expression was detectable by bioluminescence imaging already after infection with 10^5^ pfu Ad-HBV-Luc per mouse, HBeAg was only detectable at an infectious dose of 10^6^ pfu Ad-HBV-Luc (Figure 1C). In order to prove HBV-specific mRNA transcription, we analyzed the liver tissue of Ad-HBV-Luc-infected mice. Northern blot analysis revealed similar transcription levels of viral 3.2 kb and 2.4/2.1 kb mRNA after Ad-HBV-Luc and Ad-HBV1.3 infection indicating functional gene expression of HBV genome linked to the luciferase reporter (Figure 1D). We compared the secreted HBeAg and HBsAg, the latter being an HBV surface protein, levels as well as HBV genomes in the serum after infection with Ad-HBV-Luc vs. Ad-HBV1.3 (Figure 1E). We detected secreted viral antigens and HBV genomes in the serum of Ad-HBV-Luc-infected mice, although to a lesser extent than in Ad-HBV1.3-infected mice, confirming viral protein synthesis as well as the formation and secretion of HBV particles (Figure 1E). We also detected an increased number of HBV DNA copies compared to adenoviral DNA copies in the livers of infected mice, indicating HBV DNA replication (Figure 1F). These data demonstrated that Ad-HBV-Luc enables the transfer of HBV genomes into hepatocytes of mice, allowing for HBV replication, gene expression and the formation of HBV particles. Moreover, we combined Ad-HBV-Luc infection of murine hepatocytes with sensitive detection of HBV antigen load using bioluminescence imaging allowing for highly sensitive real-time monitoring of HBV infection.

### 3.2. Ad-HBV-Luc Infection Develops into Acute or Chronic HBV Infection in Mice

To evaluate the impact of viral burden on the disease outcome, we challenged mice with different viral infectious doses of Ad-HBV-Luc. We identified two doses of viral inoculum, which resulted in different kinetics of the bioluminescence signal, serum HBeAg levels and HBV pregenomic RNA (pgRNA) levels in the liver, the latter being an HBV replication intermediate. Infection with 10^8^ pfu/mouse of Ad-HBV-Luc showed stable bioluminescence signal and continuous high levels of serum HBeAg over at least 60 days after infection as well as the presence of HBV pgRNA in the liver, consistent with the establishment of a persistent infection. In contrast, infection with 10^7^ pfu/mouse of Ad-HBV-Luc showed as expected 1 log-lower initial bioluminescence signal, circulating HBeAg and HBV pgRNA, all of which were reduced to background levels after day 20 post infection (Figure 2A). Compared to viral detection by pgRNA and HBeAg, bioluminescence imaging enabled close-meshed daily measurements unraveling the onset of viral elimination of 10^7^ pfu Ad-HBV-Luc in hepatocytes. Immunohistochemical analysis of liver tissue for HBcAg expression confirmed the dose-dependent differential disease outcome at the single-cell level. Whereas we detected HBcAg-positive hepatocytes 10 days after infection in both groups, we did not detect any HBcAg-positive cells in the 10^7^ pfu/mouse group at day 30 after infection anymore (Figure 2B). Moreover, loss of HBcAg in hepatocytes, bioluminescence and circulating HBeAg was accompanied by transient elevated sALT levels during infection with the lower dose, indicating CD8 T cell-mediated anti-viral response in the liver. In contrast, high-dose infection resulted only in slightly elevated basal sALT levels without a distinct peak (Figure 2C). Altogether, these results indicate a cytotoxic immune-mediated elimination of Ad-HBV-Luc-infected hepatocytes when mice were infected with 10^7^ pfu of Ad-HBV-Luc. To verify this hypothesis, we infected mice with 10^7^ pfu of Ad-HBV-Luc and depleted CD8 T cells by means of the repeated injection of 30 µg anti-CD8α antibodies, leading to almost quantitative depletion of all CD8 T cells in the blood. This selective depletion of CD8 T cells resulted in persistent Ad-HBV-Luc infection over 60 days, determined by in vivo bioluminescence imaging, and demonstrated that the CD8 T cell population was responsible for the elimination of Ad-HBV-Luc-infected hepatocytes (Figure 2D). In summary, these data demonstrate that Ad-HBV-Luc infection can be used as a preclinical model to establish either chronic HBV infection in mice or to mount an HBV-specific CD8 T cell-mediated immune response mimicking the acute phase of HBV infection.

### 3.3. Phenotype of Endogenous HBV-Specific CD8 T Cells

We identified CD8 T cells as being responsible for eliminating Ad-HBV-Luc-infected hepatocytes and thereby clearing infection from the liver in mice. Consequently, we wondered whether HBV-specific CD8 T cells differed in their quantity or phenotype during the infection with 10^7^ pfu/mouse and 10^8^ pfu/mouse of Ad-HBV-Luc leading to either acute self-limited or chronic of infection. To identify ideal time points for CD8 T cell isolation from the livers of infected mice, we monitored anti-viral CD8 T cell immunity by bioluminescence imaging. Based on bioluminescence data, we identified day 10 as being before onset of anti-HBV immunity, and by day 30 the low dose infection had already cleared Ad-HBV-Luc infection (see Figure 2A,B). At day 15 post infection, we detected 1-log step reduction in bioluminescence, and therefore chose this day for the isolation and characterization of CD8 T cells (Figure 3A). Using MHC-I multimer-staining to identify endogenous HBV-specific CD8 T cells, we analyzed the numbers of endogenous CD8 T cells, specific for the HBcAg-derived peptide MGLKFRQL (Cor93), in the livers of mice at days 10, 15 and 30 post infection (Figure 3B). Infection with 10^7^ pfu/mouse resulted in initial expansion of endogenous Cor93-multimer-positive CD8 T cells peaking at day 15 post infection with approx. 20,000 cells per liver. The expansion of Cor93-multimer-positive CD8 T cells was followed by contraction until day 30 post infection, consistent with antigen loss in the liver and circulation (see Figure 2A,B). In contrast, during chronic HBV-infection, the numbers of Cor93-multimer-positive CD8 T cells rose at day 10 to approx. 2,000 per liver but lacked further expansion or contraction (Figure 3B). However, despite early priming of HBV-specific CD8 T cells, infection with 10^8^ pfu/mouse of Ad-HBV-Luc resulted in persistent infection. Therefore, we evaluated the phenotype of liver-associated Cor93-multimer positive CD8 T cells. During chronic HBV-infection, flow cytometric analysis revealed constant high levels of the markers PD1, TIGIT and TOX, all markers associated with T cell dysfunction. Consistently, markers associated with cytotoxicity such as GzmB and KLRG1 were elevated only at day 10 post infection and were downregulated afterwards at day 15 and 30 (Figure 3C). These results are in line with the initial fast appearance of Cor93-multimer-positive CD8 T cells but lack subsequent further expansion (Figure 3B). In contrast, Ad-HBV-Luc infection with 10^7^ pfu/mouse resulted in initial upregulation of PD1 and TIGIT on HBV-specific CD8 T cells followed by decreased expression at day 30 when the infection was cleared. We also observed overall lower levels of TOX (Figure 3C). Furthermore, GzmB and KLRG1 were upregulated during acute infection and correlated with elevated sALT levels on day 15 (Figure 3C). We further analyzed the cytokine production of CD8 T cells during the time course of Ad-HBV-Luc infection in mice at day 10, 15 and 30 post infection. To this end, we isolated liver-associated lymphocytes and restimulated these cells directly ex vivo either by Cor93 peptide or by SIINFEKL peptide as a negative control. At day 15 post Ad-HBV-Luc infection with 10^7^ pfu/mouse, during the ongoing anti-viral effector function, we detected high numbers of IFNγ and TNF producing CD8 T cells after restimulation by both Cor93 and SIINFEKL peptide, indicating that these T cells have been stimulated already at this time point during virus elimination in vivo (Figure 4A). Interestingly, CD8 T cells from 10^7^ pfu-infected mice at day 30 after virus elimination showed only TNF and IFNγ production after Cor93 peptide restimulation, but not in response to SIINFEKL peptide, hinting towards functional memory induction. In contrast, during chronic infection, we did not detect increased cytokine production upon Cor93 peptide restimulation at any time, indicating impaired functionality. Consequently, we next analyzed the effector function of total HBV-specific CD8 T cells. Therefore, we isolated CD8 T cells from livers of mice infected with either 10^7^ pfu/mouse or 10^8^ pfu/mouse Ad-HBV-Luc again at the day when bioluminescence started to drop in the 10^7^ pfu/mouse group, indicating onset of anti-viral immunity during acute disease.

Afterwards, these liver-associated CD8 T cells were co-cultured with primary murine hepatocytes infected with Ad-HBV1.3 in vitro and monitored for their specific cytotoxic effector function using impedance-based detection of hepatocyte viability (Figure 4B). The time-resolved killing assay demonstrated effector functions of liver-associated lymphocytes during 10^7^ pfu/mouse Ad-HBV-Luc infection, eliminating 80% of the hepatocytes ex vivo. Contrary to this, liver-associated lymphocytes during 10^8^ pfu/mouse infection were less functional and eliminated only 20% of the hepatocytes. Notably, using Ad-HBV1.3-infected primary murine hepatocytes as targets, we identified HBV-specific cytotoxic CD8 T cells as being responsible for the elimination of hepatocytes. Taken together, we observed the priming of HBV-specific CD8 T cells in both acute self-limited as well as chronic infection settings. During acute self-limited infection, HBV-specific CD8 T cells expanded and exhibited a functional, cytotoxic phenotype. In contrast, during chronic infection, HBV-specific CD8 T cells lacked strong expansion and showed a dysfunctional phenotype with a progressing degree of dysfunctionality from day 10 to day 30 after infection. 

### 3.4. Adoptive Transfer of HBV-Specific T Cell Receptor Transgenic CD8 T Cells to Enhance Accuracy of CD8 T Cell Phenotyping

After analysis of intrahepatic Cor93-multimer positive CD8 T cells, we wondered if the exhausted phenotype during chronic infection correlated in liver and peripheral blood. In the periphery, we were not able to detect sufficient numbers of HBV-specific CD8 T cells for phenotyping (data not shown). In order to increase the numbers of precursor HBV-specific CD8 T cells, we therefore adoptively transferred 10,000 naïve Cor93-specific CD8 T cells bearing the congenic marker CD45.1 into mice prior to infection with 10^7^ pfu or 10^8^ pfu Ad-HBV-Luc, respectively. First, we analyzed whether elevated levels of naïve Cor93-specific CD8 T cells impacted on the disease outcome. Bioluminescence imaging showed the induction of chronic HBV infection using 10^8^ pfu Ad-HBV-Luc and an expected reduction in bioluminescence in mice infected with 10^7^ pfu Ad-HBV-Luc, accompanied by a sharp peak in sALT level indicating elimination of infected hepatocytes (Figure 5A). Transfer of CD45.1 CD8 T cells did not alter the kinetics or outcome of Ad-HBV-Luc infection compared to previous experiments without adoptive T cell transfer (see Figure 2A–C).

We further characterized the expansion of CD45.1 CD8 T cells in the livers and blood of Ad-HBV-Luc-infected mice. During the drop in bioluminescence, at day 15 post infection of the acute self-limited HBV infection with 10^7^ pfu Ad-HBV-Luc, we counted approx. 45,000 CD45.1 CD8 T cells/liver and were able to detect also CD45.1 CD8 T cells in the blood (Figure 5B). Moreover, we detected CD45.1 CD8 T cells in the blood as well as in the livers of mice infected 15 days with 10^8^ pfu Ad-HBV-Luc (Figure 5B). We wondered whether the increased numbers of CD45.1 CD8 T cells in the liver during chronic infection, compared to endogenous Cor93-multimer positive CD8 T cells, were attributed to low T cell receptor expression (Figure 3B and Figure 5B). Therefore, we analyzed CD45.1 CD8 T cells for their Cor93-specific T cell receptor expression by MHC-multimer-staining during acute and chronic infection. Indeed, we observed downregulation of the Cor93-specific T cell receptor in mice infected with 10^8^ pfu Ad-HBV-Luc, compared to mice infected with 10^7^ pfu Ad-HBV-Luc (Figure 5C). These findings may explain the previously identified low numbers of endogenous Cor93-multimer-positive CD8 T cells quantified by multimer-staining of the endogenous Cor93-specific T cell receptor during chronic infection (see Figure 3B). Due to the sensitive CD45.1-staining in blood, we were able to compare the phenotype of CD45.1 CD8 T cells in the liver and peripheral compartment during acute and chronic Ad-HBV-Luc infection. To our surprise, we did not find significant differences in the expression of PD1, TOX, KLRG1 or GzmB in peripheral CD45.1 CD8 T cells (Figure 5D). Therefore, we wondered whether CD45.1 transgenic Cor93-specific CD8 T cells do not display major changes in markers attributed to exhaustion or cytotoxicity. However, CD45.1 CD8 T cells isolated from the livers of mice infected with 10^8^ pfu Ad-HBV-Luc showed still increased expression of PD1 and TOX and reduced expression of KLRG1 and GzmB, compared to their counterparts isolated from livers after 10^7^ pfu Ad-HBV-Luc infection (Figure 5E). Taken together, these data demonstrate that adoptive transfer of transgenic Cor93-specific CD8 T cells bearing the CD45.1 congenic marker does allow for sensitive detection and phenotyping of those cells after Ad-HBV-Luc infection independent of the T cell receptor down regulation. Moreover, adoptive T cell transfer does not change the outcome or kinetics of Ad-HBV-Luc infection in mice.

## 4. Discussion

Here, we describe a novel preclinical model system for HBV infection in mice applying an adenoviral vector encoding for a 1.3-overlength genome of HBV. Additionally, we added a click beetle green luciferase linked by a P2A sequence at the second truncated HBcAg open reading frame (Ad-HBV-Luc). Due to expression under the HBc promoter, we ensured the expression of the luciferase reporter protein in hepatocytes, similar to HBV proteins. Therefore, after injection of Ad-HBV-Luc into mice, we observed a distinct bioluminescence signal in the area of the liver. In addition, we detected the viral antigens HBsAg, HBeAg (the latter being a secretory form of HBcAg) as well as HBV genomes in serum of infected mice indicative for HBV particle production in the liver and their secretion into the blood. This demonstrated that Ad-HBV-Luc was capable of delivering fully functional HBV genomes into the livers of mice which can be detected by highly sensitive bioluminescence imaging in vivo.

Previously, several small animal models for hepatitis B have been developed [25,33,34]. Many of those model systems are well suited to address distinct immunological or virological questions. For example, the generation of mice carrying HBV 1.3-overlength genomes (HBV1.3-transgenic mice) enables high-level HBV replication in hepatocytes [35]. Because HBV is integrated into the genome of these mice, they display strong central tolerance to HBV antigens and HBV cannot be “cured”, therefore mimicking chronic HBV infection. Nevertheless, usage of HBV-transgenic mice has recently proved the efficacy of novel therapeutic vaccination strategies towards chronic HBV infection [27]. Additionally, another approach, transferring HBV-genomes via adeno-associated viral vectors (AAV), enabled HBV replication and led to the establishment of chronic infection in mice [36]. Interestingly, chronic hepatitis B after AAV-HBV infection is not associated with liver pathology or inflammation. Due to the establishment of chronic hepatitis B, AAV-HBV infection may also be used for the development of novel treatment strategies aiming for the elimination of chronic HBV.

However, model systems for acute self-limited and chronic HBV infection are required to identify mechanisms involved in formation of anti-HBV immunity determining either successful viral clearance or persistence. An example of a previously published mouse model of acute HBV infection is the hydrodynamic injection of HBV plasmids, which is accompanied by an initially high level of liver damage that may enable mounting an HBV-specific immunity. Another model for acute HBV infection was published using an adenovirus-based transfer of HBV genomes [28]. Thereby, infection of mice with “low” doses (10^8^ pfu/mouse) of Ad-HBV1.3 has been reported to induce chronic infection whereas a “high” dose (3 × 10^9^ pfu/mouse) of Ad-HBV1.3 induced anti-HBV immunity [37]. However, similar to hydrodynamic injection, the higher dose induced severe initial liver damage which may induce inflammation and thereby explain induction of HBV-specific immunity. Those models of acute HBV do not reflect the absent inflammation and low induction of the innate system during natural HBV infection in patients [38]. We, however, used lower infectious doses than 3 × 10^9^ pfu/mouse Ad-HBV-Luc which do not induce initial liver damage upon infection in our setting. In addition, recently, we showed that the adenovirus serotype 5 used here does not induce significant innate signaling [39]. Moreover, due to luciferase modification in Ad-HBV1.3 in our experiments, Ad-HBV-Luc was associated with lower HBV gene expression and secretion, detected by lower HBsAg and HBeAg levels in serum of mice, compared to the previously used Ad-HBV1.3. Nonetheless, with the here established Ad-HBV-Luc, the identical vector can be used to induce either acute self-limited or chronic HBV infection in mice. Similarly to Ad-HBV-Luc, it is likely that the application of a lower infectious dose of Ad-HBV1.3 results in acute self-resolved disease outcome. However, we assume that less viral particles are crucial for acute infection, because Ad-HBV1.3 revealed higher viral replication and antigen secretion in our experiments compared to Ad-HBV-Luc after infection of mice. Still, lacking a biomarker such as luciferase for real-time monitoring of HBV-infected hepatocytes will impede the detection of anti-viral immunity.

Similar observations were made in a chimpanzee study where different amounts of HBV inoculum also affected the infection outcome [40]. Thereby, chimpanzees infected with high and moderate dose of HBV controlled the infection. In contrast, low viral inoculum resulted in HBV persistence. However, the huge difference in weight and the fact that chimpanzees act as a natural host of HBV, where reinfection of new hepatocytes by viral particles takes place [40,41], makes it extremely difficult to define a common definition of an either high or low infectious dose. As it is not understood how and at which time point the viral antigen load affects the HBV immune response, it is currently difficult to state whether a high or a low dose favors viral clearance. It is also conceivable that there is a threshold in both directions. Therefore, it is difficult to directly compare the impact of the infectious dose on the disease outcome between mice and chimpanzees. Yet, here, the use of a replication-deficient adenovirus as a shuttle system and the absent infection of murine hepatocytes by HBV particles enables us to precisely analyze the effect of infectious dose on the anti-viral adaptive immunity.

The impact of the viral inoculum has recently also been demonstrated using humanized mice supplemented with human immune cells (HIS-HUHEP mice). There, the lower dose (10^7^ DNA copies) used for infection led to a lower viremia detected in the blood of mice, in contrast to infection with a high dose of infection (10^9^ DNA copies) [42]. Although there is a clear effect of the infectious dose, it is not clear which immune cells led to the control of the HBV viremia in this system, as the human immune cells and the transplanted hepatocytes were not HLA-matched. However, another model of immunodeficient mice has been repopulated with human liver progenitor cells and hematopoietic stem cells (HSC) from the same donor, allowing HLA-matched immune responses (A2/NSG-hu HSC/Hep mice) [43]. These mice have been naturally infected with HBV with doses ranging from 10^3^ to 10^7^ genome copies/mouse. Although HBV infection in these mice induced a human immune response, all mice (irrespective of the inoculum) developed chronic HBV infection. The authors demonstrated that an impaired immune response was associated with increased M2-like macrophages. In the meantime, many more valuable humanized mouse models have been developed with unique functions and outcomes of infection (mainly with chronic HBV infection) [44,45].

The detection of changes in quantity of HBV antigen load via bioluminescence imaging allows sensitive and close-meshed assessment of anti-viral immunity. This gains a huge advantage over monitoring anti-viral immunity by HBeAg measurement, which is limited to one sampling time point per week in the same animal due to the limited accessible amount of blood. Another advantage is the instantaneous response time of bioluminescence imaging. While the elimination of luciferase-expressing infected hepatocytes will immediately reduce the bioluminescence signal, circulating HBeAg will be detectable for a longer time period and the levels will decline with a certain delay. Moreover, bioluminescence imaging was 10-fold more sensitive than HBeAg measurement. However, additional expression of transgenic luciferase might also act as an antigen. Thus, firefly luciferase has been demonstrated to be immunogenic in mice [46]. However, bulk CD8 T cells isolated from 10^7^ pfu Ad-HBV-Luc-infected mice were sufficient to eliminate Ad-HBV1.3-infected hepatocytes, lacking luciferase expression. Nonetheless, we cannot formally exclude the priming of additional CD8 T cells targeting epitopes of the click beetle green luciferase used here. These results demonstrate HBV-specific T cell mediated elimination of Ad-HBV-Luc in our model system.

Using two different infectious doses, we were able to establish either acute self-limited infection (10^7^ pfu) or chronic Ad-HBV-Luc infection (10^8^ pfu). This already shows that the higher amount of expressed antigen might be responsible for CD8 T cell dysfunction. It is well known that a high antigen load correlates with CD8 T cell dysfunction/exhaustion [47,48,49,50]. After high-dose Ad-HBV-Luc infection, we also detected HBV-specific CD8 T cells with elevated levels of PD1, TIGIT and TOX, markers associated with T cell dysfunction. Those CD8 T cells also did not produce effector cytokines upon restimulation and did not perform effector function against Ad-HBV1.3-infected hepatocytes. These data may indicate that high levels of HBV antigens lead to CD8 T cell dysfunction.

During acute self-limited infection, Ad-HBV-Luc elimination occurred within a few days upon the initiation of the effector function in the liver. Quantitative analysis indicated a sharp rise in HBV-specific CD8 T cells in the liver, peaking at the days when the bioluminescence signals drops. Therefore, analysis of effector CD8 T cells requires sensitive and frequent detection of initiated immunity, which is not applicable in the measurement of serum HBeAg and HBsAg levels. Thus, using bioluminescent imaging as a tool to detect the onset of anti-viral immunity, we have been able to characterize HBV-specific effector CD8 T cells in the liver at the peak of anti-HBV immune response. Due to the low total numbers of HBV-specific CD8 T cells, we extended the Ad-HBV-Luc model by means of the adoptive transfer of naïve CD45.1 Cor93-specific CD8 T cells, increasing the precursor number [51]. Thereafter, similar to endogenous Cor93-multimer-positive CD8 T cells, transgenic CD45.1 CD8 T cells expanded and were detectable in the liver and blood of Ad-HBV-Luc-infected mice. However, CD45.1 CD8 T cells isolated from blood of chronically infected mice did not reflect the dysfunctional phenotype of their counterparts isolated from the liver. Consequently, this is in line with previously published observations that circulating HBV-specific CD8 T cells express lower levels of PD1 than their intrahepatic counterparts, indicating a less dysfunctional phenotype outside the organ of infection [22]. Thereby, the Ad-HBV-Luc mouse model reflects in part the HBV immunity described so far in patients. Caution has to be taken when analyzing PBMC-derived HBV-specific CD8 T cells, as these may not reflect numbers and phenotype of HBV-specific CD8 T cells in the liver. In conclusion, using Ad-HBV-Luc in mice enables the sensitive determination of virus elimination onset by bioluminescence imaging and thereby facilitates the analysis of momentary HBV-specific CD8 T cell response in the liver, which is distinct from the periphery.

## Figures and Tables

**Figure 1 viruses-13-02273-f001:**
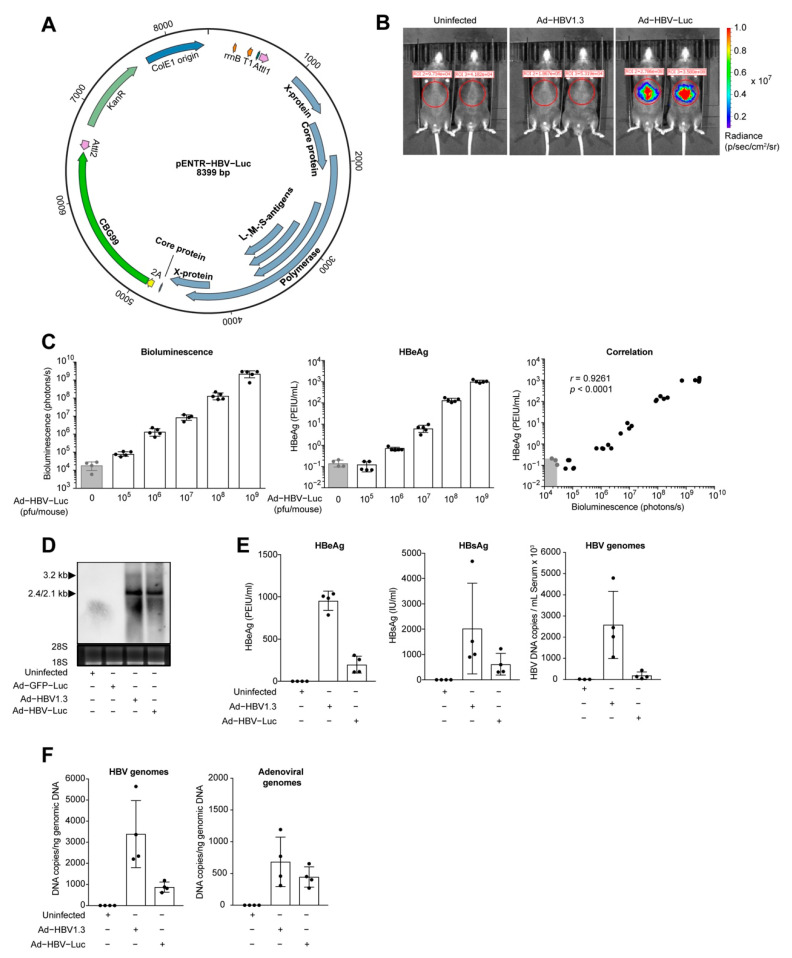
Transfer of HBV genomes in murine liver by adenoviral vectors. (**A**) Vector map of the pENTR plasmid with cloned HBV1.3 overlength genome, P2A site and CBG99 luciferase (pENTR-HBV-Luc). (**B**) Bioluminescence imaging of mice at day 8 post infection with Ad-HBV1.3 (5 × 10^8^ pfu) or Ad-HBV-Luc (5 × 10^8^ pfu). (**C**) Quantification of bioluminescence in the liver and serum HBeAg in mice infected 7 days with increasing doses of Ad-HBV-Luc (10^5^ to 10^9^ pfu). (**D**) Northern blot analysis of total liver RNA from mice infected 8 days with 5 × 10^8^ pfu Ad-GFP-Luc, Ad-HBV1.3 or Ad-HBV-Luc. 28S and 18S ribosomal RNAs served as a loading control and were detected by exposure to ultraviolet light for 0.6 s. HBV-specific fragments with 3.2 kB (pgRNA) and 2.4/2.1kb (preS/S RNA) length were detected using an HBV-specific probe. Chemiluminescence signal was recorded for 3 min. (**E**) Quantification of HBeAg, HBsAg and HBV genomes in serum of mice infected as in (**B**). (**F**) qPCR analysis of HBV genomes and adenoviral genomes in liver tissue of mice infected as in (**B**). (**A**–**F**) Mean and SD are shown (*n* = 4–5).

**Figure 2 viruses-13-02273-f002:**
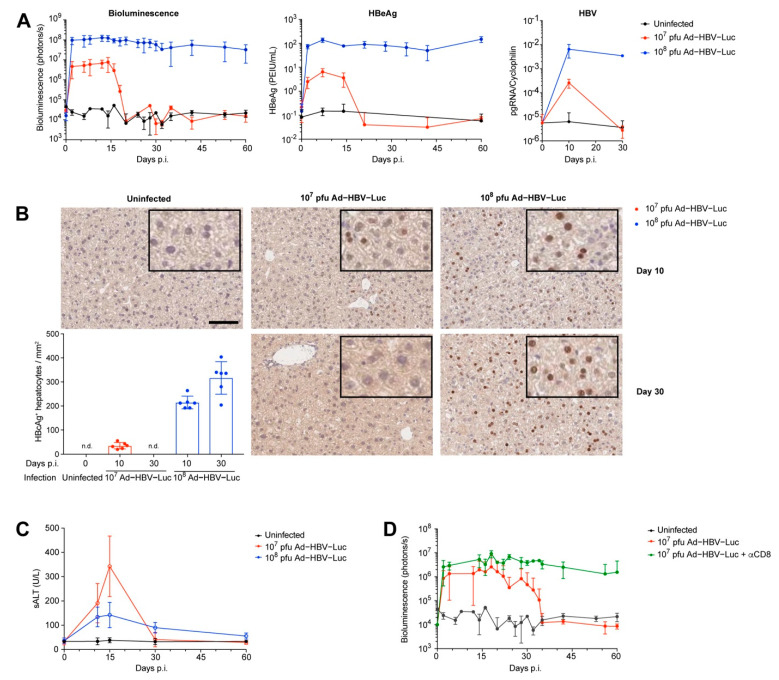
Ad-HBV-Luc infectious dose defines acute self-limited or chronic HBV infection in immunocompetent mice. (**A**–**C**) C57Bl/6 mice were infected i.v. with the acute self-limited (10^7^ pfu, red) or chronic (10^8^ pfu, blue) infectious dose of Ad-HBV-Luc. (**A**) Kinetics of the bioluminescence signal in livers, serum HBeAg levels and pgRNA in liver tissue. (**B**) Representative immunohistochemical HBcAg staining of liver sections and quantification of HBcAg-positive hepatocytes at indicated time points after infection. Scale bar = 100 µm. (**C**) Time kinetics of liver damage measured by sALT levels. (**D**) Quantification of virus-infected hepatocytes by bioluminescence signal in the liver of C57Bl/6 mice infected i.v. with the acute self-limited (10^7^ pfu) dose of Ad-HBV-Luc without (red) and with (green) treatment by 30 µg of the anti-CD8α depleting antibody. (**A**–**D**) Mean and SD are shown: (**A**) *n* = 5, (**B**) *n* = 6, (**C**) *n* = 5, (**D**) *n* = 4. All experiments were performed at least three times independently, except for (**D**).

**Figure 3 viruses-13-02273-f003:**
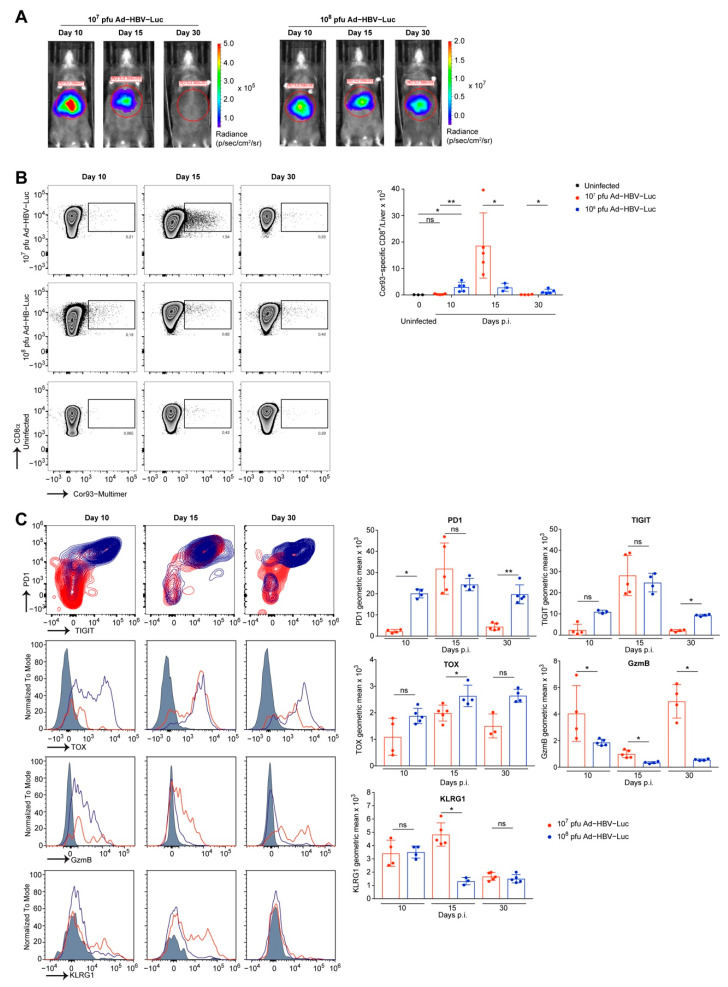
Quantity and phenotype of endogenous intrahepatic HBV-specific CD8 T cells. (**A**–**C**) Lymphocytes were isolated from livers of C57Bl/6 mice infected i.v. with the acute self-limited (10^7^ pfu, red) or chronic infectious dose (10^8^ pfu, blue) of Ad-HBV-Luc at indicated time points after infection and analyzed by flow cytometry. (**A**) Representative bioluminescence images of infected mice representing virus-infected hepatocytes. (**B**) Representative FACS plots and quantification of intrahepatic endogenous Cor93-specific CD8 T cells. Quantification in livers of uninfected mice serves as a control for unspecific staining. (**C**) Representative plots/histograms and quantification of indicated marker expression on/in Cor93-specific CD8 T cells (gated on living Cor93-specific CD8 T cells) by geometric mean. Mean and SD are shown (*n* = 3–5). The experiments were performed three times independently.

**Figure 4 viruses-13-02273-f004:**
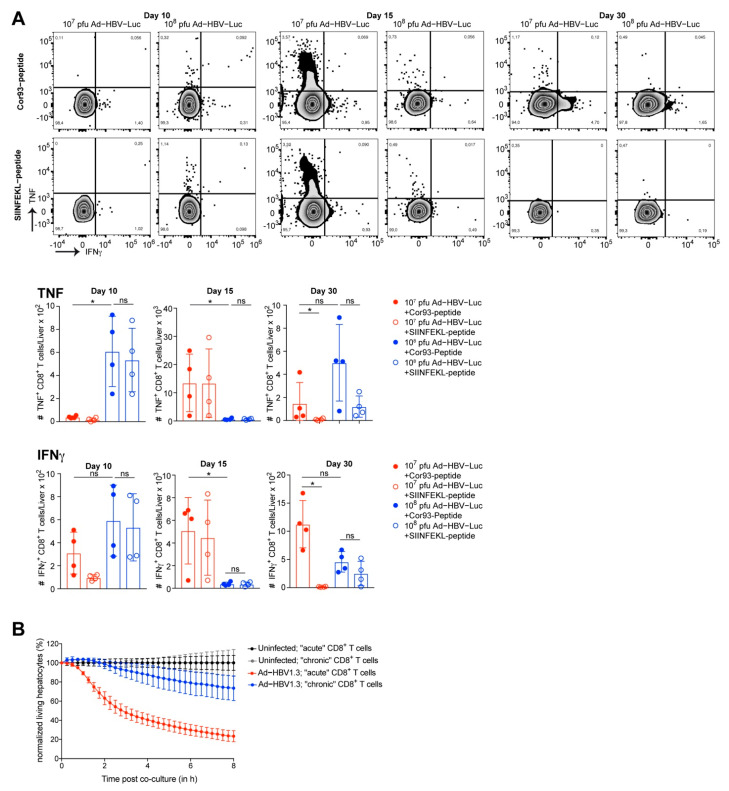
Functionality of endogenous intrahepatic HBV-specific CD8 T cells. (**A**) Lymphocytes were isolated from livers of C57Bl/6 mice infected i.v. with the acute self-limited (10^7^ pfu, red) or chronic infectious dose (10^8^ pfu, blue) of Ad-HBV-Luc. (**A**) Flow cytometric analysis and quantification of numbers of TNF and IFNγ expressing intrahepatic CD8 T cells following restimulation by HBcAg-derived Cor93 peptide or SIINFEKL peptide as control (gated on bulk living CD8 T cells) at day 10, 15 and 30 post infection. (**B**) Co-culture of primary murine hepatocytes infected with Ad-HBV1.3 and intrahepatic CD8 T cells isolated from mice infected as in (**A**) (target to effector ratio: 1 to 50). Hepatocytes were infected with MOI of 5. CD8 T cells were isolated at day 15 post infection. Lysis of target cells was detected by xCELLigence technology. Mean and SD are shown: (**A**) *n* = 4, (**B**) *n* = 5. The experiments were performed once (**A**) or twice (**B**).

**Figure 5 viruses-13-02273-f005:**
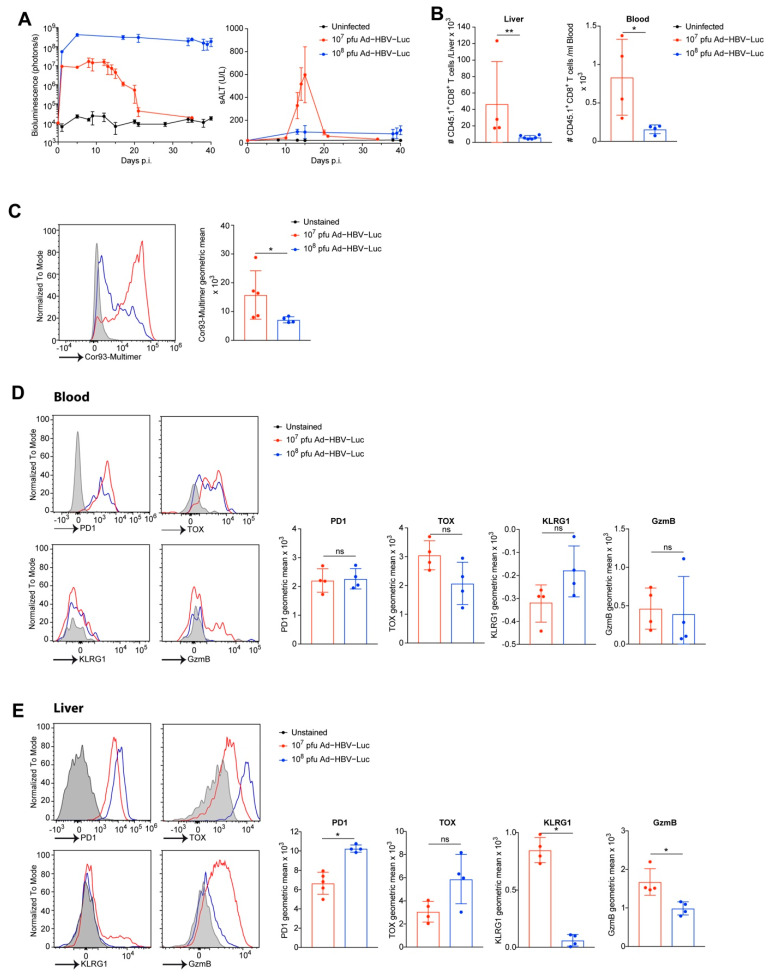
TCR transgenic CD8 T cells enhance the sensitivity of CD8 T cell phenotyping. (**A**–**E**) C57Bl/6 mice were adoptively transferred with 10^5^ naïve Cor93-specific CD8 T cells bearing the congenic marker CD45.1 and infected i.v. with acute self-limited (10^7^ pfu, red) or chronic infectious dose (10^8^ pfu, blue) of Ad-HBV-Luc. Lymphocytes from liver and blood were isolated at day 15 after infection. (**A**) Quantification of virus-infected hepatocytes by bioluminescence signal in the liver and quantification of liver damage by sALT level of infected mice at indicated time points. Detection limit was determined by measurement of uninfected mice (black). (**B**) Quantification of numbers of transferred CD45.1 CD8 T cells isolated from liver and blood. (**C**) Flow cytometric analysis of Cor93-specific T cell receptor expression on transferred CD45.1 CD8 T cells. (**D**,**E**) Representative histograms and quantification of indicated marker expression on/in transferred CD45.1 CD8 T cells (gated on living CD45.1 CD8 T cells) isolated from (**D**) blood or (**E**) liver. (**A**–**E**) Mean and SD are shown (*n* = 5). The experiments were performed two times independently.

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
