# Peer review of "In Vivo Bioluminescence Imaging of HBV Replicating Hepatocytes Allows for the Monitoring of Anti-Viral Immunity"

_viruses, 2021, doi:10.3390/v13112273_

Round 1

Reviewer 1 Report

Manske et al. has developed an in vivo model with an HBV luciferase reporter to study anti-viral immunity. The authors demonstrate that a recombinant adenoviral vector expressing an HBV-luciferase genome is able to induce acute and chronic HBV infection and the associated CD8 T cell responses. This tool could be interesting for the study of HBV-specific immune responses and is worth publishing in Viruses. Some minor points need however to be addressed.

Specific comments:

Lane 99: Change « teil » to « tail »

Lane 129:  The use of cyclophilin PCR is not explained, was it used for normalization?

Lane 188: The sequences of probes used to detect HBV should be shown

Paragraph 2.10: Please add a sentence on the use of 28S and 18S ribosomal RNAs and the probes sequences for their detection

Paragraph 2.2 Flow cytometry: detail the antibody dilutions for cell staining and show antibody specificity by a staining of a cell line negative for the target proteins

Lane 292: The quantity of anti-CD8α per mice for CD8 T cell depletion needs to be indicated

Figure 2: The efficiency of the CD8 T cell depletion should be indicated

Reviewer 2 Report

The research article by Manske et al., describes the development of a recombinant adenovirus expressing a HBV 1.3-overlength genome linked to a luciferase reporter. This tool provides a detection method for identification of infected hepatocytes and monitoring the development of immune responses against the virus in mice. Interestingly, the authors observed that the initial infectious dose was a key factor determining the development of either acute of chronic HBV infection. Moreover, the authors identified a dysfunctional T CD8 response as one of the underlying mechanisms related to HBV persistence in their animal model.

The manuscript is very well written, describing the results in a clear and coherent manner. In regards to the methodology, the reviewer considers that the work includes the necessary controls and validation experiments to support the notions put forward by the authors. Moreover, such a tool could indeed be quite practical for the study of HBV infection in vivo. The reviewer suggests only a few minor corrections and modifications.

Specific comments:

Minor comments:

  • Taking into account a previous report employing HIS-HUHEP mice (PMID: 28851562), it could be argued that the sentence in page 2 lines 72-74 should be down toned. The authors of the previously mentioned work also found differences in infection outcome depending on the initial viral dose employed. Moreover, it was also found that low or high doses had an impact on CD8 T cell responses. Therefore, the reviewer considers that this work should be at least cited in the discussion.
  • There seems to be a problem with the sequence provided for the reverse primer targeting HBV pgRNA, as it is the same as the reverse primer targeting cyclophilin (page 3, line 128).
  • The authors should decide if they add a space before the percentage symbol or not, and stick to it throughout the manuscript (e.g., page 3, lines 148 and 192).
  • Replace overlenght by overlength (page 14, line 449 and 462).
  • Replace natrium chloride by sodium chloride (page 2, line 93).
  • Replace Alanin by Alanine (page 3, line 107).
  • Replace primer by primers (page 3, line 119).
  • Replace IFNg by IFNγ (page 12, line 368).
  • Point missing at the end of the paragraph (page 5, line 220).

Reviewer 3 Report

In this manuscript, Manske et al., prepared a recombinant adenovirus expressing HBV genome linked to luciferase to analyse an HBV-specific immune response in mice. The experiments are performed well and largely support their conclusions. I have several comments and questions that should be addressed before publication.

I have several suggestions: 

  1. Figure 1A needs better resolution. You can also improve the vector description in methods.
  2. On line 265 you stated: we combined HBV infection of murine hepatocytes. I would rather use a term Ad-HBV infection, because this is not a typical HBV infection.
  3. The second paragraph of discussion (line 460-473) does not add any important idea or information concerning your results. I also think that the general idea of this paragraph is partially included in the introduction (line 62-67). What I personally miss in the discussion, is a comparison with some humanized HBV-mouse models (Bility et al., PloS Pathog. 2014, Yuan et al., Hepatology 2019).

I also have several questions:

  1. Adenovirus serotype 5 can be highly immunogenic compared to HBV (missing IFN-response etc.). Did you measure the induction of innate immunity especially IFN-I/III production or upregulation of interferon-stimulated genes after Ad-HBV-luc infection in mice? 
  2. In Figure 1C, HBeAg correlated with the infectious dose of Ad-HBV-Luc. Did you observe the same correlation also for HBsAg? Do you have any explanation why the control virus (Ad-HBV1.3) was more effective in the production of HBeAg and HBV genomes than Ad-HBV-Luc even though the amount of adenoviral DNA copies in liver tissue was quite similar for both Ad-constructs (figure 1E + F)?
  3. For several systems using adenovirus expressing HBV or AAV-HBV, high amount of HBV replicative intermediates (dsDNA) or even cccDNA formation were reported. Did you analyse cccDNA formation after Ad-HBV-Luc infection in mice? Is there any sign of HBV dsDNA integration in murine hepatocytes after Ad-HBV-Luc infection?
  4. In the paper Rinker et al., J Hepatol. 2018, authors described that HBsAg loss was corelated with the improvement of exhaustion phenotype of T lymphocytes. Did you analyse the seroconversion towards HBsAg or different levels of HBsAg during low dose (107) or high dose (108) Ad-HBV-luc infection in mice?
